# Towards Novel Fluorinated Methacrylic Coatings for Cultural Heritage: A Combined Polymers and Surfaces Chemistry Study

**DOI:** 10.3390/polym11071190

**Published:** 2019-07-16

**Authors:** Valentina Sabatini, Eleonora Pargoletti, Valeria Comite, Marco Aldo Ortenzi, Paola Fermo, Davide Gulotta, Giuseppe Cappelletti

**Affiliations:** 1Dipartimento di Chimica, Università degli Studi di Milano, Via Golgi 19, 20133 Milano, Italy; 2Consorzio Interuniversitario per la Scienza e Tecnologia dei Materiali (INSTM), Via Giusti 9, 50121 Firenze, Italy; 3CRC Materiali Polimerici “LaMPo”, Dipartimento di Chimica, Università degli Studi di Milano, Via Golgi 19, 20133 Milano, Italy; 4The Getty Conservation Institute, 1200 Getty Center Drive, Suite 700, Los Angeles, CA 90049, USA; 5Dipartimento di Chimica, Materiali e Ingegneria Chimica “Giulio Natta”, Politecnico di Milano, Via Mancinelli 7, 20131 Milano, Italy

**Keywords:** fluorinated methacrylic polymer, Botticino marble protection, aging, photo-chemical stability, outdoor exposure

## Abstract

In this work, new co- and ter-polymers of methyl methacrylate (MMA), ethyl methacrylate (EMA), and *N*-butyl methacrylate (nBuMA), containing just 1% mol × mol^−1^ of a fluorinated co-monomer, 3,3,4,4,5,5,6,6,7,7,8,8,8-tridecafluoro-octyl methacrylate (POMA), were synthesized. After an UV accelerated aging test, the photo-chemical stability of the polymers prepared was determined by ^1^H NMR and FT-IR spectroscopy, size exclusion chromatography, differential scanning calorimetry and wettability measurements. The polymers were applied to Botticino tiles to achieve better performances in terms of water repellency and consequently deterioration resistance. One-year prolonged exposure to a real environment was conducted and the properties of the coated materials and their performances were studied using different surface techniques such as water contact angle (WCA) and colorimetric measurements (CIELaB), capillary absorption, permeability (RVP) tests and soluble salts determination. The effectiveness of the fluorinated methacrylic coatings was clearly demonstrated; among all the resins, the co-polymer MMA_POMA seems to be the most performing one. Furthermore, both the UV photo-chemical resistance and the easiness of removal was successfully studied.

## 1. Introduction

The cultural heritage deterioration is a spontaneous and irreversible decay process, common to a wide number of stone-based building materials [1]. Particularly, water percolation into stone substrates and air pollutants deposition onto their surfaces are reputed the main responsible of stones degradation [2], especially in the case of carbonatic materials [3] such as Botticino marble [4], largely used in the ancient architecture and modern art [5]. Once water and harsh agents enter the stone, several physico-chemical processes, e.g., carbonate dissolution [6], freezing/thawing cycles [7] and/or crystallization/precipitation [8], surface soiling [9] can happen, dramatically affecting the quality and properties of such building materials.

Thus, the application of protective coatings to stone materials is the key to success in order to preserve stone artefacts from decay, limiting the water penetration into stone porous substrates. Any stone protective coating must satisfy several requirements, such as good adhesion and water repellency, thus hindering the absorption of harsh agents into the stones and concomitantly favouring the natural water vapor permeability. It should be also transparent, easy to apply and remove, without ruining the original substrates surface, as well as durable [10,11,12].

Among the different typologies of materials that are considered traditionally as stone protectives, polyacrylates [13] play a key role, thanks to their broad range of useful features, such as transparency, easy film-forming capability, good mechanical properties, cheap and flexible synthesis processes [14,15]. In particular, in the last decade acrylic-based materials were widely used as protective coatings in the field of the conservation of ancient architectures and modern buildings [16]. However, such kind of materials have serious drawbacks, e.g., poor durability, scarce hydrophobicity and water vapor permeability [17].

To overcome such drawbacks, for many years partially fluorinated acrylic polymers have been extensively studied from both an academic and industrial point of view. Specifically, (*i*) Bernett et al. [18] conducted a study about the molar proportions between fluorinated and not fluorinated polymer constituents, (*ii*) Imae [19] synthesized novel homopolymers which were fluorinated in main, terminal and side chains respectively and (*iii*), Panayiotou et al. [20] presented a comprehensive study about the surface properties of different fluorinated methacrylic foils. Furthermore, such kinds of resins have been widely studied in recent years to assess their photo-chemical stability and water repellency when employed as outdoor coating materials [21]. Despite the improvement in terms of hydrophobic properties, the same drawbacks described for polyacrylates remain unresolved [22]. Thus, there is still a pressing need to develop high performance protectives able to overcome the technical problems previously reported for polyacrylates and their fluorinated derivatives.

Accordingly, it is well known that the replacement of hydrogen atoms, deriving from acrylic monomers, in alpha position to carbonyl groups with alkyl groups, e.g., -CH_3_ groups, improves the photo-chemical and thermal stability of the resulting resins [23]. In this scenario, the features of new fluorinated methacrylic polymers were deeply investigated, with the goal of conjugating hydrophobic properties, deriving from fluorine-based co-monomers, with the intrinsic photo-chemical stability of polymethylmethacrylates [24,25,26,27].

Hence, the present work deals with the synthesis, via free radical polymerization, and characterization of co- and ter-polymers obtained using different combinations of methyl methacrylate (MMA), ethyl methacrylate (EMA), *N*-butyl methacrylate (nBuMA) and a fluorinated methacrylic monomer, 3,3,4,4,5,5,6,6,7,7,8,8,8-tridecafluoro-octyl methacrylate (POMA), as well as the evaluation of the performance of these resins when used as protective coatings for stones. The work was aimed at obtaining polymer materials with improved photo-chemical stability, good film-forming properties, easiness of application, water repellency and good water vapor permeability, minimizing the fluorine content due to the high costs related to the synthesis and the use of fluorinated compounds, i.e., just at 1% mol × mol^−1^. In this context, a multi-disciplinary approach based on analyses techniques typical of polymers and surfaces chemistry was used to characterize resins, either pristine or applied on Botticino stones. An accelerated UV aging test to evaluate the photo-chemical stability of these new materials was carried out. Modifications of the chemical composition, monitored via ^1^H NMR and FT-IR spectroscopy, molecular weights, molecular weights distributions, thermal and wetting properties were studied to determine the possible presence and the nature of the photo-degradation processes. Once applied on Botticino substrates, the durability and the effect of aging caused by the prolonged exposure (about 12 months) to a real environment was also studied. In particular, the three synthesized resins were applied by paint-coating onto Botticino tiles using a low environmental impact solvent, i.e., ethyl acetate, and our attention was devoted to the analysis of wetting, colorimetric, vapor permeability/capillarity absorption behaviors and salts crystallization resistance, comparing the aged fluorinated coatings to the unaged ones. Finally, the possibility of coatings removal was studied via ^1^H NMR and FT-IR spectroscopy, size exclusion chromatography (SEC) and thermal analyses on the dissolved resins and using water contact angle, water vapor permeability and capillarity measurements on the cleaned marbles, in order both to study the coatings photo-chemical stability and to assess the resins easiness of removal.

## 2. Materials and Methods

### 2.1. Materials

Methyl methacrylate (MMA, 99%), ethyl methacrylate (EMA, 99%), *N*-butyl methacrylate (nBuMA, 99%), methacryloyl chloride (97%), 1H,1H,2H,2H-Perfluoro-1-octanol (97%), α,α’-Azoisobutyronitrile (AIBN, 99%), triethyl amine (TEA, ≥99.9%), sodium bicarbonate (NaHCO_3_, ≥99.7%), sodium sulfate (Na_2_SO_4_, ≥99.99%), cyclohexane (99.5% anhydrous), methanol (99.8% anhydrous), ethyl acetate (≥99.9%), distilled water Chromasolv^®^ (≥99.9%), tetrahydrofuran (THF, ≥99.9 anhydrous), dichloromethane (DCM, ≥99.8 anhydrous), hydrochloric acid (HCl, 37%) and chloroform-d (CDCl_3_, 99.96 atom % D) were supplied by Sigma Aldrich (Milan, Italy) and used without purification. The lithotype used, purchased from Marmi Orobici Graniti s.p.a. (Bergamo, Italy), is an almost pure micritic limestone, predominantly white-light brown in color, dolomitized in a negligible quantity, with limited porosity. It belongs to the upper portion of the Corna Formation, known commercially as Botticino Classico Marble, present on the south-western slope of Monte Fratta, in the Municipality of Botticino (Brescia), placed in Northern Italy. This limestone has been used in Brescia since Roman times (Tempio di Vespasiano, Theater, Foro, Terme); subsequently it was used, in the same city, for the construction of different buildings (such as Church of Santa Maria dei Miracoli, Palazzo della Loggia, and Duomo Nuovo). At the end of the nineteenth century, it was also used for the realization of the monument of the Vittoriano in Rome [28]. The characterization of the stone (mineralogical, chemical and porosimetric) was performed in a previous work by Fermo et al. [29].

### 2.2. Synthesis of MMA_POMA, MMA_nBuMA_POMA and EMA_nBuMA_POMA

The 3,3,4,4,5,5,6,6,7,7,8,8,8-tridecafluoro-octyl methacrylate (POMA) monomer was synthesized and characterized in accordance with the synthetic procedure reported in our previous works [24,25]. A co-polymer, MMA_POMA, and two ter-polymers, MMA_nBuMA_POMA and EMA_nBuMA_POMA, were synthesized via free radical polymerization. Figure 1 shows the reaction scheme together with the exact amount of the monomers used for the syntheses. In a typical polymerization procedure, a 250 cm^3^ one-necked round bottom flask was equipped with a reflux condenser having a nitrogen inlet adapter and an overhead magnetic stirrer. The flask was flushed with nitrogen, charged with 40 cm^3^ of cyclohexane and the following feeds were used, according to the polymer desired: (i) MMA_POMA co-polymer: MMA, POMA (mol_POMA_ = 1% of mol_MMA_) and AIBN (mol_AIBN_ = 1% of mol_MMA+POMA_), the latter used as free radical initiator; ii) MMA_nBuMA_POMA ter-polymer: MMA, nBuMA (mol_MMA_ = mol_nBuMA_), POMA (mol_POMA_ = 1% of mol_MMA+nBuMA_) and AIBN (mol_AIBN_ = 1% of mol_MMA+nBuMA+POMA_) and iii) EMA_nBuMA_POMA ter-polymer: EMA, nBuMA (mol_EMA_ = mol_nBuMA_), POMA (mol_POMA_ = 1% of mol_EMA+nBuMA_) and AIBN (mol_AIBN_ = 1% of mol_EMA+nBuMA+POMA_). The polymerization mixture was put in an oil bath, heated for 24 h at 70 °C and then gradually cooled to room temperature. The solution obtained was precipitated into a large excess of methanol under stirring and a white solid precipitate was obtained. The product was recovered via filtration and, given the difficulties in completely removing unreacted monomers, the polymers were washed with fresh methanol for several days under stirring. After washing, they were dried in a vacuum oven (about 4 mbar) at 40 °C for 48 h and then the absence of residual solvents was checked via the isothermal thermogravimetric analysis (TGA Perkin Elmer Italia, Milan, Italy), performed for 2 h at 70 °C under nitrogen flow.

### 2.3. Polymer Foils Preparation and Relative Characterization before and after UV Test

Polymer foils were obtained by depositing in a 7 × 7 × 1 cm PTFE mould a solution of 20% w/w of the polymer itself in ethyl acetate. Then, the solvent was evaporated for 24 h in a vacuum oven (about 4 mbar) at 40 °C. The foils thickness, evaluated by the Nikon eclipse ME600 optical microscope (equipped with the Nikon digital camera light DS_Fi1, software Nis-Elementi BR, magnification 50×, Nikon Corporation, Tokyo, Japan), was in the range of 200–250 µm. To evaluate the stability of the polymers under UV radiations, an accelerated aging test was performed according to the UNI 10925:2001 standard method [30]. The test was conducted for 100 h at room temperature and ambient pressure by means of an Ultra Vitalux lamp (315–400 nm of UVA rays and 280-315 nm of UVB ones with effective power densities of 4.5 and 14.0 mW cm^−2^, respectively).

Wetting properties of both the unaged and aged foils were assessed by water contact angle (WCA) measurements using a KRÜSS Easydrop Instrument (KRÜSS GmbH, Hamburg, Germany). WCA values were obtained by depositing a drop (around 5 μL) of Milli-Q water. All the data were averaged over at least ten measurements on each sample to guarantee a statistical population.

^1^H NMR spectra were collected at 25 °C with a BRUKER 400 MHz spectrometer (Bruker Italia Srl, Milan, Italy). All samples were prepared by dissolving 8–10 mg of polymer in 1 cm^3^ of CDCl_3_.

Fourier Transform-Infrared (FT-IR) spectra were obtained on a Spectrum 100 spectrophotometer (Perkin Elmer Italia, Milan, Italy) in attenuated total reflection (ATR) mode using a resolution of 4.0 and 256 scans, in a range of wavenumber between 4000 and 400 cm^−1^. A single-bounce diamond crystal was used with an incidence angle of 45°. FT-IR spectra of the samples obtained after the UV aging test were also collected.

The effect of UV irradiation on the molecular weights properties of the synthesized polymers was evaluated using a size exclusion chromatography (SEC) system having Waters 1515 Isocratic HPLC pump (Waters Spa, Milan, Italy) and a four Waters Styragel (10^3^Å-10^4^Å-10^5^Å-500Å) columns set with a RI detector Waters 2487 using a flow rate of 1 cm^3^ × min^−1^ and 40 μL as an injection volume. Samples were prepared dissolving 40 mg of polymer in 1 cm^3^ of anhydrous THF; before the analysis, the solution was filtered with 0.45 μm filter. Molecular weight data were expressed in polystyrene (PS) equivalents. The calibration was built using monodispersed PS standards having the following nominal peak molecular weight (Mp) and molecular weight distribution (D): Mp = 1,600,000 Da (D ≤ 1.13), Mp = 1,150,000 Da (D ≤ 1.09), Mp = 900,000 Da (D ≤ 1.06), Mp = 400,000 Da (D ≤ 1.06), Mp = 200,000 Da (D ≤ 1.05), Mp = 90,000 Da (D ≤ 1.04), Mp = 50,400 Da (D = 1.03), Mp = 37,000 Da (D = 1.02), Mp = 17,800 Da (D = 1.03), Mp = 6520 Da (D = 1.03), Mp = 5460 Da (D = 1.03), Mp = 2950 Da (D = 1.06), Mp = 2032 Da (D = 1.06), Mp = 1241 Da (D = 1.07), Mp = 906 Da (D = 1.12); ethyl benzene (molecular weight = 106 g mol^−1^). For all analyses, 1,2-dichlorobenzene was used as an internal reference.

Differential scanning calorimetry (DSC) analyses were conducted using a Mettler-Toledo DSC1 (Mettler-Toledo S.p.A., Milan, Italy); the analyses were conducted weighting 5–10 mg of each sample in a standard 40 µL aluminium pan, heating from 25 °C to 125 °C at 10 °C min^−1^ and using an empty 40 µL aluminium pan as a reference. The thermal behaviour of the samples obtained after the UV aging test was studied using the same thermal program.

### 2.4. Application of Polymer Coatings onto Botticino Marble and External Exposure

Eight Botticino tiles of 5 × 5 × 1 cm (two uncoated as references) and thirty-two of 2 × 2 × 2 cm (eight used as references) were dried in an oven at 60 °C for a week. Twenty percent *w*/*w* of polymers was dissolved in ethyl acetate. The application of coatings on tiles surfaces was carried out using a brush with bristles made with Polyamide 6 to obtain a homogeneous and thin polymer layer of around 200–250 μm. Thickness and surface morphology of the uncoated and coated tiles were characterized by profilometry, using a Bruker DektakXT (Bruker Italia Srl, Milan, Italy) contact profilometer in according with the procedure described in our previous work [24]. As shown in Figure 2, bare and coated stones were exposed in a typical urban environment placed in Milan, the Duomo Cathedral terrace (site II: Central balcony, 19.7 m above the ground), that is in the city center, a representative site of a typical urban polluted environment. The exposure test was carried out for 345 days (20 April 2018 to 31 March 2019). All specimens were exposed with a horizontal orientation. For the present exposure, a total of 20 samples (four of 5 × 5 × 1 cm; sixteen of 2 × 2 × 2 cm) were used; the remaining samples are still exposed and will be studied in future works. The wetting properties of the bare and coated samples were determined before and after the exposure test, in accordance with the procedure described in Section 2.4.

Colorimetric measurements were performed to verify the colour modification of the protective films after the exposure test. The chromatic coordinates were calculated according to the Commission Internationale d’Eclairage (CIELaB method) [31], starting from diffuse reflectance spectra acquired in the UV-Vis spectral range from 800 to 300 nm with a UV-Vis SHIMADZU UV 2600 spectrophotometer (SHIMADZU, Milan, Italy). The global colour differences (ΔE*) of marbles, induced upon coating application, is calculated in accordance to Equation (1).
(1)$$\Delta E^{*}=\sqrt{\Delta L^{*2}+\Delta a^{*2}+\Delta b^{*2}}$$
where L*, a* and b* are the lightness (0 for black-100 for white), the red-green (positive for red and negative for green) and the yellow-blue (positive for yellow and negative for blue) components, respectively. According to the literature, no significant variation occurs when ΔE* < 5 [10].

Capillary water absorption analyses were performed on bare and coated materials, before and after the exposure test, by the gravimetric sorption technique, as described in the standard protocol UNI EN 15,801 [32]. In particular, the amount of the absorbed water per unit area (Q_i_) is calculated as reported in Equation (2).
(2)$$Q_{i}=m_{i}-m_{0}/A$$
where m_**i**_ is the weight of the sample in contact with water at a specific time t_**i**_, m_**0**_ is the initial weight of the sample, and A is the sample area in contact with water during the test. The total amount of water absorbed by a material at the end of the test (Q_ft_) and the capillary absorption (CA, i.e., the slope of the capillary curve within the first 30 min of the tests) parameters were determined, accordingly.

Water vapour permeability (WVP) analyses of bare and coated mortars were evaluated by means of the methodology described in the standard protocol UNI 15,803 [33]. The reduction of water vapour permeability (RVP) was calculated according to Equation (3).
(3)$$\%\mathrm{RVP}=\left(\frac{m_{f,\;\mathrm{untreated}}-m_{f,\;\mathrm{treated}}}{m_{f,\;\mathrm{untreated}}}\right)\cdot 100$$
where m_**f,untreated**_ and m_f,treated_ are the weight of bare and the treated sample at the end of the test. It is reported that a polymer material cannot be used as a protective for mineral substrate if RVP is higher than 50% [27]. WVP analyses were repeated on materials subjected to the above-mentioned external exposure.

Substrates porosity was evaluated by means of the water saturation method (WSM). This methodology is based on the determination of the open pore volume V_op_ (cm^3^) which corresponds to the volume of water absorbed by the samples, according to the procedure reported in the European protocol (UNI EN 2001) [34]. The total open porosity (TOP) is calculated as the volume of water penetrated within the sample divided by the geometric volume of the sample itself.

Ion chromatography (IC) was employed for the quantification of the main inorganic constituents of the deposits. About 2 mg of polymer powder, collected in accordance with our previous work [35], was placed in a test tube and treated with 10 mL of Milli-Q water. The solutions were put in an ultrasonic bath for 1 h, then centrifuged for 30 min and injected for IC analyses. Measurements of anionic (NO_3_^−^ and SO_4_^2−^) species were carried out by using an ICS-1000 HPLC system equipped with a conductivity system detector. The analysis was carried out with an IonPac AS14A column (Dionex S.p.a., San Donato Milanese, Italy) using 8 mM Na_2_CO_3_/1 mM NaHCO_3_, flow rate = 1 cm^3^ × min^−1^, using a conductivity system detector working with an anion self-regenerating suppressor ULTRA (ASRS-ULTRA) (Dionex S.p.a., San Donato Milanese, Italy) [12,36]. A blank sample consisting of a marble specimen, both not treated and not exposed, was also analyzed (namely absolute white). Anions concentrations determined in the blank sample were subtracted from values obtained from the exposed ones, either bare or treated.

### 2.5. Coatings Removal from Botticino Samples and Further Polymers and Stones Characterization

5 × 5 × 1 cm coated Botticino tiles were immersed in 20 cm^3^ of ethyl acetate for 30 min, washed several times with fresh ethyl acetate and dried in an oven at 60 °C for 24 h. Then, on the cleaned-up tiles capillary water absorption, WVP and WCA tests were performed as described in the previous paragraphs. On the other side, after solvent evaporation in a vacuum oven (around 4 mbar) at 40 °C for 24 h, the removed coatings were analyzed by ^1^H NMR and FT-IR spectroscopy, SEC and DSC analyses. The samples were prepared and characterized in accordance with Section 2.3.

## 3. Results and Discussion

### 3.1. Synthesis of MMA_POMA, MMA_nBuMA_POMA and EMA_nBuMA_POMA Resins and Their Characterization before and after the UV Aging Test

Primarily, the structure of the synthesized polymers (MMA_POMA, MMA_nBuMA_POMA and EMA_nBuMA_POMA) was confirmed by ^1^H NMR spectroscopy, as reported in Figure 3.

The real amount of POMA in the samples was quantitatively determined by ^1^H NMR spectra using Equation (4) where “I_POMA_” is the integral area of peak [a] (highlighted in blue in Figure 3), “I_MMA_” is the integral area of peak [b] (highlighted in red in Figure 3), “2” corresponds to the number of POMA protons related to the integral area of peak [a] and “3” is the number of MMA protons for the integral area of peak [b].
(4)$$\mathrm{POMA}\;\%\;\mathrm{mol}\times {\mathrm{mol}}^{-1}=\frac{\left(I_{POMA}/2\right)}{\left(I_{MMA}/3\right)}$$


The experimental values obtained, 0.99 mol × mol^−1^ for MMA_POMA, 0.98 mol × mol^−1^ in the case of MMA_nBuMA_POMA and 0.99 mol × mol^−1^ for EMA_nBuMA_POMA, are fully in accordance with the theoretical one, confirming that all the fluorinated monomers were successfully inserted in the methacrylic matrix.

The long-term stability of the new tailored fluorinated resins was determined via an UV accelerated aging test. Figure 3 shows a comparison between ^1^H NMR spectra collected before and after the aging test: (a) and (b) for MMA_POMA samples, (c) and (d) in the case of MMA_nBuMA_POMA resins and (e) and (f) for EMA_nBuMA_POMA polymers, respectively. In the case of aged samples, the variation of the macromolecular structure and their relative POMA loading before and after the aging test is negligible (0.99 mol × mol^−1^ for MMA_POMA, 0.98 mol × mol^−1^ in the case of MMA_nBuMA_POMA and 0.98 mol × mol^−1^ for EMA_nBuMA_POMA), clearly evidencing the high photo-chemical stability of the synthesized resins.

Appendix A shows a comparison between FT-IR spectra of the samples collected before and after the UV aging test. Only slight differences between the unaged and aged nBuMA-based samples are appreciable and limited only to the side chains chemical bonds; specifically no significant variations in the C–H aliphatic bonds of the main chains occur [37,38]. In general, for polyacrylates materials, the shape of the bands related to C–H aliphatic bonds, the carbonyl groups and the same C–O groups conjugated to carbonyl esters, change dramatically after the UV aging test, due to the presence of hydrogen atoms, deriving from acrylic monomers, in alpha position to carbonyl groups that are able to start the photo-chemical degradation of the polymers themselves [39]. Here, the –CH_3_ groups, deriving from methacrylic monomers, in alpha position to carbonyl functionalities are not susceptible of polymer bonds degradation phenomena, resulting in resins with noteworthy photo-chemical stability [40].

Furthermore, the effect of UV rays was determined by the evaluation of the number average molecular weight ($\bar{Mn}$) and the molecular weight distribution (D) obtained by SEC analyses (Table 1). Fluorinated resins remain soluble in THF, i.e., the solvent adopted for SEC measurements, after the aging test, suggesting the absence of radicals that allow the formation of a partially reticulated polymer via cross-linking reactions [41]. Furthermore, the photo-chemical stability of the tailored resins is clearly evidenced by the slight variation of the molecular weights and their relative distribution before and after the aging test.

The thermal properties of the resins were evaluated (Table 1, 5th column). The T_g_ values of the three samples are respectively 89, 62 and 48 °C. Thus, by increasing the length of the aliphatic pendant chains in the co-monomers EMA and nBuMA, the T_g_ tends to decrease. After the aging test, T_g_s remain almost the same.

The wetting properties of the polymer foils were investigated by WCA measurements (Table 1, 6^th^ column) on both the air and mold sides, evidencing a higher hydrophobic degree in the case of the latter ones. In accordance with our previous works [24,25,26,42], the WCA difference between the air and PTFE sides for all the polymer foils (around 20°) can be ascribable to the re-organization of the fluorinated groups of the polymer chains during the evaporation of the solvent and to the high affinity with the hydrophobic PTFE mold surface. Notably, all the polymer substrates show similar wettability features. Furthermore, the aging test does not affect the polymer surface properties, corroborating the previous results concerning the variation of both molecular weights and T_g_s.

### 3.2. Characterization of Botticino Coated Samples and Exposure in a Real Polluted Environment

Notwithstanding the surface features of the fluorinated acrylic polymers are desired and fully exploitable in the field of stone protectives, the poor solubility of fluoro-groups in common organic and inorganic solvents [43] dramatically limits the coating formulation. However, in this case, once the polymers were synthesized and characterized, all the coatings were easily dissolved in ethyl acetate, thanks to the very low fluorine content in the resulting polymers, i.e., just the 1% mol × mol^−1^, and deposited on the Botticino substrates.

To assess the film thicknesses and their relative roughness, profilometry analyses were performed. By scratching the surface substrates with a sharp pin, the layers height was evaluated. Figure 4 shows either the thickness (δ) or the roughness (<R>) values for the MMA_POMA-treated marble (MMA_POMA@B), as a representative sample. We managed to obtain a very good reproducibility by paint-brushing the investigated protective agents, achieving ca. 250 μm thick films with an average roughness of around 20–25 μm, for all the tested clad-materials. Moreover, an increase of the surface roughness was observed with respect to the bare marble, whose value is about (7 ± 1) μm, helping to enhance the surface hydrophobicity of the coated marbles (see the following section).

#### 3.2.1. Surface Properties, Water Capillarity and Vapour Permeability of Treated Marble

It is well known that the application of POMA-based coatings can allow obtaining of hydrophobic surfaces in the case of cultural heritage buildings and monuments [24,25]. Hence, experimental WCA measurements were carried out to evaluate the wettability features of both bare and treated Botticino marbles. Notably, the pristine substrate shows an almost hydrophilic character (θ < 90°; Table 2, 2nd column); whereas, after the treatment with the as-synthesized water-repellent agents (MMA_POMA@B, MMA_nBuMA_POMA@B and EMA_nBuMA_POMA@B), an increase of the surface hydrophobicity occurs (θ around 80–90°; Table 2, 2^nd^ column). In particular, among the investigated resins, the copolymer MMA_POMA seems to guarantee the highest hydrophobic behavior. Moreover, all the protective coatings were found to be transparent (ΔE* < 5; Table 2, 4th column) without an appreciable marble colour variation after the application of the hydrophobic layers.

As widely stated in the literature [31,44,45], the hindering of water absorption by capillarity plays a pivotal role in the cultural heritage preservation. Indeed, water, permeating through the materials pores, causes either cracks due to the freezing-melting cycles or salts crusts formation, owing to the transport of these substances inside the substrates. Alongside a complete block of water capillary rinse, the water vapor transmission rate through the pores must remain almost unchanged after the coating treatments, assuring the proper vapor regime inside the materials [11,46]. The reduction of vapor permeability can be considered negligible up to 50% as stated in the literature [47]. Therefore, a promising protective agent should drastically reduce the water absorbed by capillarity, allowing at the same time the proper vapor permeability.

Regarding the former property, the amount of water absorbed per unit area over time (Q_i_) was plotted with respect to the square root time (s^1/2^) for all the studied samples (Figure 5a). As Figure 5a displays, the application of all the three modified-POMA resins can lead to a sharp decrease of water absorbed, with respect to the bare marble. Specifically, the most performing coating seems to be EMA_nBuMA_POMA. Moreover, from the graphs, Q_ft_ and CA parameters were calculated. Additionally in this case, a drastic reduction of these two parameters is observable for all the treated tiles (Table 2, 6th and 8th columns). Particularly, CA collapses of about two orders of magnitude, after coatings application, confirming the high hydrophobicity of the investigated protective agents. Furthermore, by comparing the present data with literature, similar values were obtained. For instance, Aslanidou et al. [48] deeply investigated the protective features of an emulsion composed by alkoxy silanes and organic fluoropolymer (namely Silres BS29A), applied on both sandstones and marble specimens. Notably, the reported Q_*ft*_ values (i.e., around 20 and 6–8 mg × cm^−2^, respectively for bare and coated marbles) are very close to the ones shown here. Actually, in this study, a reduction of ca. 70–80% of water absorbed by capillarity is achieved, similarly to Silres-treated samples. In particular, even in relation to literature data [48,49], the EMA_nBUMA_POMA sample results in the most performing coating, showing a very drastic decrease of water absorbed. Furthermore, to corroborate the previous results, porosity evaluation was carried out by means of the water saturation method [34]. After the hydrophobic treatment, an important decrease of the marble porosity was observed with respect to the bare sample (% TOP of 1.30%), especially in the case of MMA_nBuMA_POMA@B (i.e., 0.29%, whereas for MMA_POMA@B and EMA_nBuMA_POMA@B are 0.62% and 0.40%, respectively).

On the other hand, as concern for the vapor permeability, Figure 5b shows the cumulative mass change per unit area for each set of successive weighing versus time. Notably, MMA_POMA@B and MMA_nBuMA_POMA@B show a good behavior, similar to the bare Botticino marble, leading to a small decrease (about 25%; Table 2, 10th column) in material transpiration. This is very promising, since the protective agents allow stones to breath, reducing at the same time the deterioration phenomena, caused by external agents such as atmospheric pollution. Vice versa, EMA_nBuMA_POMA polymer revealed to be less efficient in guaranteeing a good level of transpiration, since the RPV percentage is very close to the threshold value of 50%. Hence, also in the case of vapor permeability, the comparison with literature data, about fluorine-based polymers [48,50], underlines the goodness of the obtained results, since %RPV values are quite similar to the ones presented herein.

#### 3.2.2. Real External Exposure Test

To evaluate the effect of atmospheric pollution on cultural heritage materials, an external exposure test was carried out [11,31,51]. The aim of this test is to assess the effectiveness of the applied protective coatings in the prevention or, at least, in the reduction of stone deterioration. Hence, an exposure test in a real polluted environment was performed, exposing both bare and treated Botticino marbles on the Duomo Cathedral terrace (see Figure 2). After almost one year-exposure, colorimetric, capillary water absorption, WVP and WCA measurements were carried out. Table 2 reports all the results obtained after the exposure period. Concerning the surfaces wettability, the bare sample has a slightly higher hydrophobic behavior, that is probably connected to the presence of carbonaceous dirt related to the outdoor environment (Table 2, 2nd and 3rd column). On the contrary, MMA_nBuMA_POMA@B and EMA_nBuMA_POMA@B maintain almost the same hydrophobicity (θ around 80°), whereas MMA_POMA@B slightly loses its starting features (θ from 90° to 77°; Table 2, 3rd column). Moreover, no yellowing effect, typical of aging processes [16], occurred for the clad-marbles (Table 2, 5th column); only in the case of EMA_nBuMA_POMA an appreciable colour variation was observed (ΔE* > 5). Moreover, the water capillary absorption and WVP were investigated again. Regarding the former, very similar values of Q_ft_ and CA were achieved, with the exception of the final water amount absorbed by MMA_nBuMA_POMA@B, that registered a drastic absorption reduction (9.8 versus 1.3 mg cm^−2^). This may be related to some carbonaceous dust/dirt presence on the substrates surface, which hindered the water capillary rise. Interestingly, the percentage of RVP remains quite the same for MMA_POMA@B and MMA_nBuMA_POMA@B, whereas it slightly increases for the other ter-polymer, going beyond the threshold limit of 50% [47].

Hence, by comparing these findings with the data obtained after the accelerated UV aging tests on pure polymers, it is possible to infer that both the co-polymer MMA_POMA and the ter-polymer MMA_nBuMA_POMA are more stable than the EMA_nBuMA_POMA, remaining undamaged even after a prolonged outdoor exposure. Conversely, the EMA-based ter-polymer preserved its physical features to the detriment of the chemical ones, as observed by means of the UV aging tests (Section 3.1).

Furthermore, the interaction among atmospheric gases and aerosol with stone surfaces is one of the main environmental degradation processes that occur on the surfaces of monuments and historical buildings, exposed in polluted environments [52,53]. Particularly, sulphates are the products of carbonatic stones degradation (due to SO_2_ oxidation and subsequent interaction with the stone itself), whereas nitrates are mainly caused by atmospheric deposition [12,36,54,55,56,57,58,59].

Hence, herein, main anions were analyzed in the one-year exposed marble samples. The obtained average concentrations are shown in Table 3. Specifically, the B-pure sample SO_4_^2−^ concentrations are comparable with the already reported data, in the case of Carrara marble; conversely, NO_3_^−^ concentrations are six-time higher, probably due to some dissolution phenomena occurred on the stone surfaces, depending on different meteorological conditions [57]. By comparing sulphate and nitrate concentrations in the bare specimens with the concentrations in the aerosol particulate matter, it can be observed that these values are in accordance with the typical ones, already found in Milan [36,54,55,56,57,59].

### 3.3. Coatings Removal from Botticino Samples: Polymers and Stones Characterization

After the exposure test, all the three coatings were removed from Botticino tiles through several washing steps with ethyl acetate. Hence, the macromolecular and thermal properties of the aged resins were analyzed via ^1^H NMR, FT-IR, SEC and DSC analyses to check their photo-chemical stability. The capillary absorption and air-breathing behavior of the cleaned-up Botticino tiles were evaluated to demonstrate the easiness of the protective removal, allowing the pristine stone features. Figure 6 shows MMA_POMA (a) and MMA_nBuMA_POMA (b) resins do not ruin the marble surfaces as seen after coatings removal; on the contrary, in the case of EMA_nBuMA_POMA (c) a black insoluble layer is visible on the Botticino surface.

Thus, to hypothesize the possible polymer degradation process, aliquots of the insoluble coating were analyzed by FT-IR spectroscopy (Figure 6d), but no significant variation between the soluble and insoluble fractions of the coating removed is detectable. However, the changing solubility of the EMA_nBuMA_POMA coating appears as the prevailing pathway of crosslinking reactions [24,39,60]. Indeed, according to Chiantore et al. [61], the completely different behavior already described for MMA-based polymers and the EMA_nBuMA_POMA resin seems to suggest that the light stability of these materials decreases with the presence of not fluorinated methacrylate co-monomers following the sequence: Short chain n-alkyl > long chain n-alkyl. The instability to light of polymers with long chain n-alkyl-methacrylic esters is in agreement with the fact that the methyne hydrogen of acrylate/methacrylate –CH_2_ units can be involved into photo-degradation mechanisms, i.e., the longer the –CH_2_ alkyl chains, the more unstable to the light the acrylic/methacrylic polymer [62].

On the other side, comparing unaged MMA_POMA and MMA_nBuMA_POMA ^1^H NMR spectra (Figure 3) and FT-IR curves (Appendix A) to the ones obtained after removal (Appendix A), no significant differences are detectable. Moreover, regarding SEC analyses (Figure 7), no variation of the molecular weights and their relative distribution occurs for MMA_POMA resin (Figure 7a). On the contrary, a small weight loss was obtained in the case of the MMA_nBuMA_POMA sample (Figure 7b). As stated in the literature, both the reduction of the average molecular weight without a great variation of D value (Figure 7c) and no evidences of cross-linking reactions (i.e., the sample does not change its solubility before and after the exposure test) are the prevailing outcomes of photo-degradative de-polymerization processes, in terms of chains fragmentation [61]. However, considering the length of the external exposure test and the very small difference of the MMA_nBuMA_POMA molecular weights before and after the exposure test, the latter resin can be considered a protective coating with satisfactory shelf-life.

A further corroboration of the present findings was obtained by evaluating the wettability, capillary absorption and vapor permeability of the Botticino tiles cleaned from MMA_POMA and MMA_nBuMA_POMA (Figure 6d). By comparing all the results obtained by means of these three analyses, we can infer that both the marble features are restored, and the coatings can be completely and easily removed.

Lastly, in terms of thermal data (Figure 7c), T_g_s remain almost the same for both samples.

## 4. Conclusions

In this work, methyl methacrylate (MMA), N-butyl methacrylate (nBuMA) and ethyl methacrylate (EMA) co- and ter-polymers were synthesized via free radical polymerization using just 1% mol × mol^−1^ of a fluorinated methacrylic monomer, 1H,1H,2H,2H-Perfluoro-octyl-methacrylate (POMA). Thus, a multi-disciplinary approach based on analyses techniques typical of polymers and surfaces chemistry was used to characterize resins, either pristine or applied on Botticino marble.

At first, the high photo-chemical stability of POMA-based resins was exploited by means of an accelerated UV aging test, showing very promising results.

Afterwards, the synthesized protectives were applied by paint-coating on Botticino substrates and their long-term properties caused by the one year-outdoor exposure were also studied. All the three methacrylic-based fluorinated polymers showed very promising features as protective agents. Notably, the co-polymer seems to be the most suitable one, due to its ability to preserve either the vapor permeability or limited water absorbed by capillarity without occluding the marble pores. Conversely, for the two ter-polymers, especially in the case of MMA_nBuMA_POMA, the pores clogging was further corroborated by ion chromatography since a nitrate ions accumulation was observed on the Botticino surface.

Finally, the resins photo-chemical stability and easiness of removal from Botticino tiles were investigated as fundamental parameters. Additionally, in this case, the co-polymer reveals to be the optimal one, since no marble appearance variation was noticed after coating removal and the polymer itself remained stable from a physico-chemical point of view.

Hence, to the authors’ best knowledge the multi-disciplinary approach adopted in this work has not been reported so far. Specifically, the present research has led to a deep comprehension of the macromolecular and surface properties of methacrylic-based fluorinated polymers, that can be successfully adopted as protectives for the cultural heritage.

## Figures and Tables

**Figure 1 polymers-11-01190-f001:**
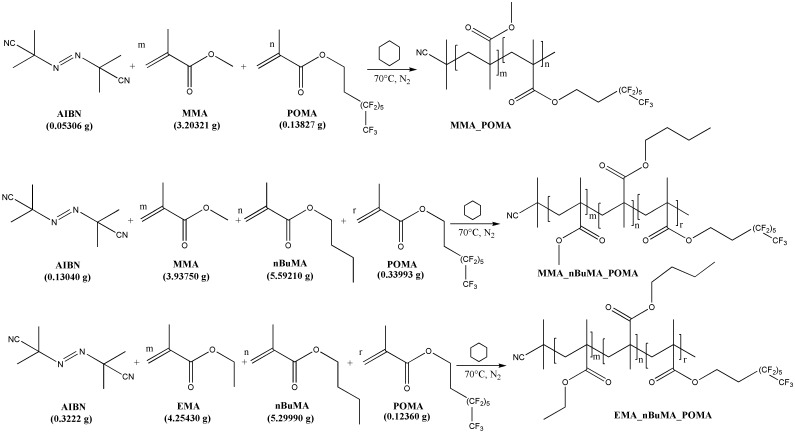
Synthetic route and loading of the reagents for all the synthesized polymers.

**Figure 2 polymers-11-01190-f002:**
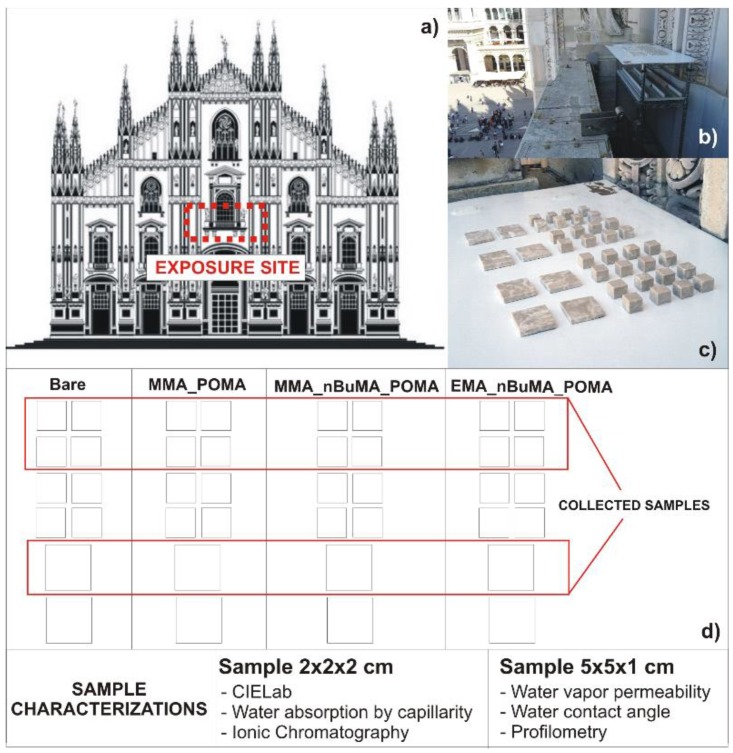
Schematics of the exposure site indicating (**a**) the location; (**b**,**c**) the samples collected and (**d**) the methods used for the characterization of the samples.

**Figure 3 polymers-11-01190-f003:**
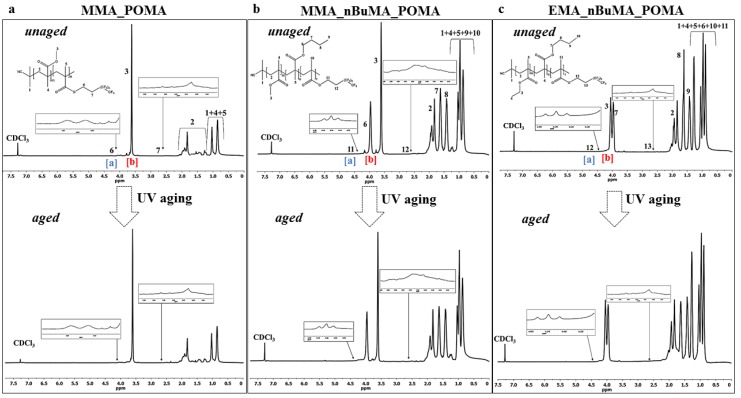
^1^H-NMR (proton nuclear magnetic resonance) spectra of both unaged and ultra violet (UV) aged (**a**) MMA_POMA (methyl methacrylate_3,3,4,4,5,5,6,6,7,7,8,8,8-tridecafluoro-octyl methacrylate, (**b**) MMA_nBuMA_POMA (methyl methacrylate_ *N*-butyl methacrylate _3,3,4,4,5,5,6,6,7,7,8,8,8-tridecafluoro-octyl methacrylate) and (**c**) EMA_nBuMA_POMA samples (ethyl methacrylate_ *N*-butyl methacrylate _3,3,4,4,5,5,6,6,7,7,8,8,8-tridecafluoro-octyl methacrylate).

**Figure 4 polymers-11-01190-f004:**
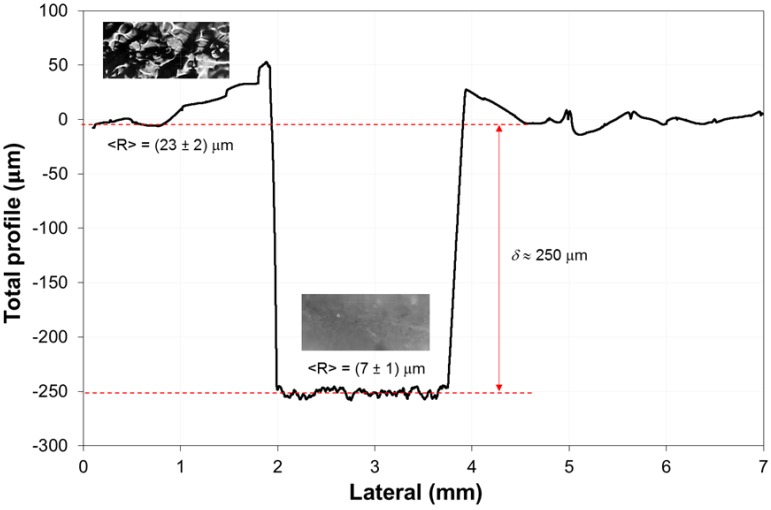
Topographic profile of MMA_POMA-treated marble (MMA_POMA@B), as a representative sample. Roughness data are reported as average values with the relative standard deviations. The coatings thickness (highlighted by red arrows) were evaluated by scratching the films with a pin.

**Figure 5 polymers-11-01190-f005:**
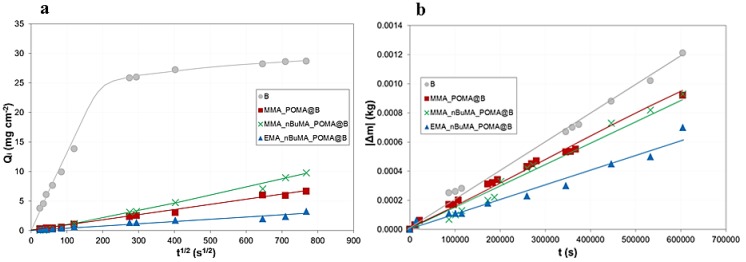
(**a**) Water capillary absorption and (**b**) water vapor permeability tests by differently POMA-clad Botticino substrates.

**Figure 6 polymers-11-01190-f006:**
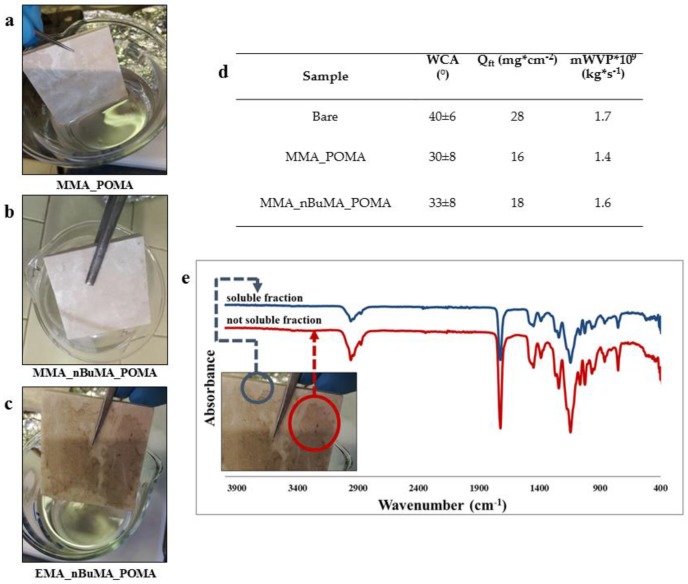
Cleaned marble tiles coated by (**a**) MMA_POMA, (**b**) MMA_nBuMA_POMA, (**c**) EMA_nBuMA_POMA resins and (**d**) table showing WCA, the final amount of water absorbed by capillarity, slopes of the vapor permeability curves measured on cleaned-up Botticino tiles. About EMA_nBuMA_POMA coating removed after the external exposure: (**e**) FT-IR spectra of both soluble and insoluble coating fractions.

**Figure 7 polymers-11-01190-f007:**
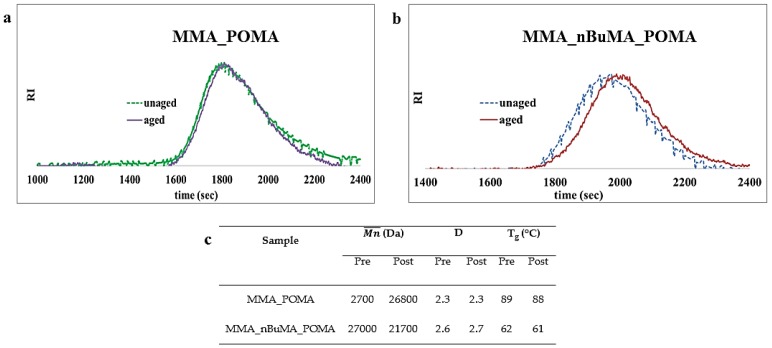
SEC curves of (**a**) MMA_POMA and (**b**) MMA_nBuMA_POMA resins before and after the external exposure test; (**c**) comparison of MMA_POMA and MMA_nBuMA_POMA molecular weights and thermal data before and after the exposure.

**Table 1 polymers-11-01190-t001:** Size exclusion chromatography (SEC), thermal and water contact angle (WCA) data of the resins synthesized collected before and after the UV aging test.

Samples	$\bar{Mn}$ (Da)		D		T_g_ (°C)		θ (°) Air-Side		θ (°) Mold-Side	
	UV Aging									
	Pre	Post	Pre	Post	Pre	Post	Pre	Post	Pre	Post
MMA_POMA	27000	25000	2.3	2.4	89	87	72 ± 2	74 ± 3	93 ± 1	91 ± 1
MMA_nBuMA_POMA	27000	22100	2.6	2.7	62	64	75 ± 1	73 ± 2	94 ± 2	94 ± 3
EMA_nBuMA_POMA	26800	23400	2.6	2.8	48	47	69 ± 2	72 ± 4	96 ± 4	92 ± 4

**Table 2 polymers-11-01190-t002:** Comparison of WCA, colorimetric variation assessment (ΔE*), water capillary absorption by both the final quantity of water absorbed (Q_ft_) and the capillary absorption (CA) coefficient, reduction percentage of vapor permeability (% RVP) relative to bare and coated marbles, before and after the outdoor exposure.

Samples	WCA (°)		ΔE*_(Treated-Bare)_		Q_ft_ (mg × cm^−2^)		CA (mg × cm^−2^ × s^−1/2^)		% RVP	
	Outdoor Exposure									
	Pre	Post	Pre	Post	Pre	Post	Pre	Post	Pre	Post
B	40 ± 6	70 ± 7	−	−	28.7	32.7	0.143	0.102	−	−
MMA_POMA@B	90 ± 3	77 ± 7	3.2	3.1	6.6	5.6	0.003	0.005	24	30
MMA_nBuMA_POMA@B	82 ± 1	83 ± 4	3.0	2.8	9.8	1.3	0.005	0.005	23	25
EMA_nBuMA_POMA@B	82 ± 2	80 ± 4	1.4	7.1	3.3	4.2	0.010	0.016	42	53

**Table 3 polymers-11-01190-t003:** Average concentrations of sulphate and nitrate anions determined on the B substrates exposed at the terrace of the Duomo cathedral during the one-year exposure.

Samples	Anions (µg × cm^−2^)	
	NO_3_^−^	SO_4_^2−^
B	20 ± 5	24 ± 8
MMA_POMA@B	21 ± 5	7 ± 4
MMA_nBuMA_POMA@B	44 ± 7	2 ± 1
EMA_nBuMA_POMA@B	19 ± 4	9 ± 1

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
