# Peer review of "Towards Novel Fluorinated Methacrylic Coatings for Cultural Heritage: A Combined Polymers and Surfaces Chemistry Study"

_polymers, 2019, doi:10.3390/polym11071190_

Round 1
Reviewer 1 Report
Fluorinated methacrylic coatings which could be used for the protection of natural stone are synthesized and characterized. Moreover, the efficacy of the produced coatings to protect Botticino tiles is evaluated. This is a very carefully designed and implemented study: New materials are synthesized. A large variety of characterisation techniques are employed. The effects of the artificially accelerated and natural ageing are studied. I believe that the manuscript can be published in Polymers. There are some minor concerns, however, which can be addressed by the authors to improve their manuscript and increase its potential impact, as described next.
Figures included in the Supplementary file (e.g. Figure 1S) should not be discussed in the main text. Only the conclusions drawn by the figures should be included in the main manuscript and the detailed discussion should be included in the Supplementary file.
A paragraph should be added in the Introduction describing previous works on synthesis and characterization of fluorinated acrylic polymers. For example: European Polymer Journal,Volume 46, Issue 2, 2010, Pages 202-208 and others.
In principle, major drawbacks of fluorinated acrylic polymers are their low solubility in typical solvents and their high costs. The products of the authors dissolve in ethyl acetate which is good. It would be useful to discuss these issues.
The molecular weights of the produced polymers are low. Is there a reason for choosing low molecular weights for the experiments? Are the polymers entangled?
Is contact profilometry suitable to scan “soft” polymer surfaces? What was the force applied by the instrument onto the polymer surface?
Comparison of the reported results with previously published results would increase the impact of the work. For example, how are the results of Figure 5 (water capillary absorption and water vapor permeability tests) compared with corresponding results achieved with other polymer coatings suggested for stone protection? e.g. Materials 2018, 11, 585; doi:10.3390/ma11040585 and other.
Equations used to calculate Qi, % RVP, Delta“m” (and any other quantity included in the manuscript) should be described. This would be useful particularly for the readers of Polymers who are not necessarily experts in stone conservation.
I think that the “Conclusions” should be rewritten. Only conclusions should be included in the corresponding paragraph. General statements and description of the implemented work should not be included in the “Conclusions” paragraph.
In ref 43 “A.” should be capital.
Author Response
Response to Reviewer 1 Comments
Thank you for the Reviewer’s constructive comments that contribute to improve the overall quality of our manuscript. We have taken into consideration all the suggested remarks. The answers to reviewer comments will appear in red in the following text.
Reviewer 1 Comments:
Fluorinated methacrylic coatings which could be used for the protection of natural stone are synthesized and characterized. Moreover, the efficacy of the produced coatings to protect Botticino tiles is evaluated. This is a very carefully designed and implemented study: New materials are synthesized. A large variety of characterization techniques are employed. The effects of the artificially accelerated and natural ageing are studied. I believe that the manuscript can be published in Polymers. There are some minor concerns, however, which can be addressed by the authors to improve their manuscript and increase its potential impact, as described next.
Point 1: Figures included in the Supplementary file (e.g. Figure 1S) should not be discussed in the main text. Only the conclusions drawn by the figures should be included in the main manuscript and the detailed discussion should be included in the Supplementary file.
Response 1: In according to the Reviewer’s suggestion, a detailed description of supplementary Figures has been moved from the main text to the Supporting Section (lines 23-30 and 36-39) maintaining the Figure comments in the main manuscript (lines 276-285).
Point 2: A paragraph should be added in the Introduction describing previous works on synthesis and characterization of fluorinated acrylic polymers. For example: European Polymer Journal, Volume 46, Issue 2, 2010, Pages 202-208 and others.
Response 2: In the Introduction section (lines 59-65), we have added a paragraph reporting recent and relevant literature (J. Phys. Chem. 1962, 66, 1207; Curr. Opin. Colloid Interface Sci, 2003, 8, 307; Eur. Polym. J. 2010, 46, 202) about the synthesis of novel fluorinated acrylic polymers and their surface characterizations.
Point 3: In principle, major drawbacks of fluorinated acrylic polymers are their low solubility in typical solvents and their high costs. The products of the authors dissolve in ethyl acetate which is good. It would be useful to discuss these issues.
Response 3: Thank you for the Reviewer’s constructive observation that contributes to improve the overall quality of the work. We have added some comments (lines 93-94 and 316-321) about fluoropolymers processing in different organic and inorganic solvents, underlining the “green” properties of the adopted solvent, ethyl acetate.
Point 4: The molecular weights of the produced polymers are low. Is there a reason for choosing low molecular weights for the experiments? Are the polymers entangled?
Response 4: All the fluoropolymers have been synthesized via a free radical polymerization technique, without any control over the mechanism of polymer chains growth and macromolecular weights. This leads to fluorinated acrylic polymers characterized by low molecular weights and broad molecular weights distribution. However, on the other side, we obtain polymers having very good processability and easy film forming capability. Furthermore, the present technique is quite versatile and easy to apply in an industrial scale-up process. Lastly, checking the solubility and rheological properties of these samples, no entangled structure formation is detected.
Point 5: Is contact profilometry suitable to scan “soft” polymer surfaces? What was the force applied by the instrument onto the polymer surface?
Response 5: Profilometry measurements can be performed on polymeric and organic surfaces, as well. However, in order to avoid any samples damages, these analyses were carried out by setting up n-lite mode, using a sub-micron tip of 12.5 mm and stylus force of about 3 mg, which is low enough to assure the correctness of the measurement, without ruining the polymeric films.
Point 6: Comparison of the reported results with previously published results would increase the impact of the work. For example, how are the results of Figure 5 (water capillary absorption and water vapor permeability tests) compared with corresponding results achieved with other polymer coatings suggested for stone protection? e.g. Materials 2018, 11, 585; doi:10.3390/ma11040585 and other.
Response 6: According to the Reviewer’s comment, a detailed comparison with literature data has been made. Particularly, three new references (Materials 2018, 11, 585; Journal of Cultural Heritage 3 (2002) 309–316; Coatings 2018, 8, 429) along with the relative comments have been added into the Results and Discussion paragraph (lines 370-378 and 391-394).
Point 7: Equations used to calculate Qi, % RVP, “Dm” (and any other quantity included in the manuscript) should be described. This would be useful particularly for the readers of Polymers who are not necessarily experts in stone conservation.
Response 7: As suggested by the Reviewer, a detailed explanation of the equations for the determination of water capillary absorption parameters, reduction of water vapor permeability and global color variation (DE*) have been added into the manuscript (see Materials and Methods paragraph, lines 205-222).
Point 8: I think that the “Conclusions” should be rewritten. Only conclusions should be included in the corresponding paragraph. General statements and description of the implemented work should not be included in the “Conclusions” paragraph.
Response 8: In accordance with the Reviewer’s remark, the Conclusions section (lines 488-520) has been completely rewritten.
Point 9: In ref 43 “A.” should be capital.
Response 9: We have corrected ref. 43, accordingly.
Reviewer 2 Report
The paper entitled "Towards novel fluorinated methacrylic coatings for 2 Cultural Heritage: a combined Polymers and Surfaces 3 Chemistry study" is well written and reports on timely findings on the fluorinated methacrylic coatings. These polymers are technically versatile in many applications and have been demonstrated to be stable. The methods employed are state-of-the-art and clearly presented. The quality of the figures is high and the discussion of results interesting to a broad auditory. Therefore, I recommend its publication as it stands.
Author Response
Response to Reviewer 2 Comments
Dear Reviewer 2, we have been honored to read your opinion about our research. Thank you for the time and attention given to our work.
Reviewer 2 Comments:
The paper entitled "Towards novel fluorinated methacrylic coatings for Cultural Heritage: a combined Polymers and Surfaces Chemistry study" is well written and reports on timely findings on the fluorinated methacrylic coatings. These polymers are technically versatile in many applications and have been demonstrated to be stable. The methods employed are state-of-the-art and clearly presented. The quality of the figures is high and the discussion of results interesting to a broad auditory. Therefore, I recommend its publication as it stands.